# Analyzing the COVID-19 Transmission Dynamics in Acre, Brazil: An Ecological Study

Joseane Elza Tonussi Mendes [1,2], Blanca Elena Guerrero Daboin [1,2], Tassiane Cristina Morais [3,4], Italla Maria Pinheiro Bezerra [3], Matheus Paiva Emidio Cavalcanti [1,2,5], Andres Ricardo Perez Riera [1], Matias Noll [6] and Luiz Carlos de Abreu [1,2,4,5,*]

[1] Laboratory of Studies Design and Scientific Writing, Postgraduate Division, University Center FMABC, Santo André 09060-870, SP, Brazil; tonussidoutorado2018@gmail.com (J.E.T.M.); bgdaboin@yahoo.com (B.E.G.D.); mpaivaemidio@gmail.com (M.P.E.C.); andres.riera@gmail.com (A.R.P.R.)
[2] School of Medicine, University of Limerick, V94 T9PX Limerick, Ireland
[3] School of Sciences of Santa Casa de Misericórdia de Vitória (EMESCAM), Vitoria 29045-402, ES, Brazil; morais.tassiane@gmail.com (T.C.M.); italla.bezerra@emescam.br (I.M.P.B.)
[4] Department of Integrated Health Education, Federal University of Espirito Santo, Vitoria 29075-910, ES, Brazil
[5] Post-Graduate Program in Medical Sciences, Faculty of Medicine, University of São Paulo, São Paulo 01246-903, SP, Brazil
[6] Department of Education, Instituto Federal Goiano, Ceres 76300-000, GO, Brazil; matias.noll@ifgoiano.edu.br
* Correspondence: luizcarlos@usp.br

**Abstract:** The north region of Brazil is characterized by significant vulnerabilities, notably surpassing national poverty indicators. These disparities exacerbated the impact of respiratory illnesses on the healthcare system during the COVID-19 pandemic, particularly in areas with limited healthcare resources, inadequate infrastructure, and barriers to healthcare access. The crisis was further influenced by multiple lineages that emerged as significant virus variants associated with increased transmissibility. Within this context, our ecological study focused on analyzing the epidemiological evolution of COVID-19 in the state of Acre. We constructed time-series trends in incidence, lethality, and mortality from March 2020 to December 2022 using the Prais–Winsten regression model. Our findings revealed that in 2020, there was an increasing trend in incidence, while mortality and lethality continued to decrease ($p < 0.05$). In the following year, both incidence and mortality decreased, while lethality increased at a rate of 1.02% per day. By the end of 2022, trends remained stationary across all rates. These results underscore the importance of ongoing surveillance and adaptive public health measures to bolster the resilience of healthcare systems in remote and vulnerable regions. Indeed, continuous monitoring of the most predominant SARS-CoV-2 lineages and their dynamics is imperative. Such proactive actions are essential for addressing emerging challenges and ensuring effective responses to adverse situations.

**Keywords:** COVID-19; pandemic; epidemiology; incidence; mortality; lethality; Amazonia; Brazil





## 1. Introduction

The COVID-19 pandemic, sparked by the SARS-CoV-2 virus, a Betacoronavirus [1], emerged in December 2019 in Wuhan, China, and swiftly evolved into a global health and economic crisis. By 2020, the World Health Organization (WHO) had declared it a pandemic and urged nations to implement containment measures due to a surge in global cases and fatalities [2,3]. As of the first case until 28 February 2024, the worldwide tally was 774,699,366 confirmed COVID-19 cases and 7,033,430 deaths. Although the WHO lifted the public health emergency declaration in May 2023, COVID-19 remains a significant global threat. Even four years after the start of the pandemic, countries such as the United States of America, China, India, France, Germany, and Brazil reported the highest numbers of accumulated cases, with the United States of America and Brazil also recording the most deaths during the same period [4].

Given the ongoing global situation, we must enhance our understanding of the clinical and epidemiological aspects of SARS-CoV-2 to develop new diagnostic, therapeutic, and healthcare management strategies. This is particularly crucial for Brazil, which has experienced a disproportionately high number of cases and fatalities compared to other nations [4]. Until 5 March 2024, there have been 38,521,738 confirmed COVID-19 cases and 710,174 deaths; however, the distribution of new cases varied across different regions of the country [5].

The northern Brazilian region is characterized by widespread poverty, with all states exhibiting poverty indicators surpassing the national average [6–8] and Acre ranking among the poorest states in the country [8]. These disparities have exacerbated the impact of respiratory illnesses on the healthcare system during the COVID-19 pandemic, particularly in areas facing limited healthcare resources [8,9] and discernible discrepancies in healthcare infrastructure and investment relative to other regions of Brazil [10]. Many municipalities in Acre also face geographic isolation due to inadequate transportation and communication networks, with approximately 59% categorized as rural [11].

Acre also boasts significant natural resources [12], with its vast forest cover accounting for 87% of its land. This forest cover has been vulnerable to fires, mainly during the dry season [13], contributing to increased particulate matter (PM2.5) pollution that adversely affects ecosystem health [14]. The potential link between short-term and long-term exposure to PM2.5 and COVID-19 has significant implications for public health in Brazil, where incidence and mortality rates are among the highest worldwide [15]. Particularly socially disadvantaged communities, including indigenous people, bear the brunt of high exposure levels, with mortality rates from COVID-19 surpassing the national average [16]. Under these circumstances, monitoring epidemiological indicators in these regions and creating interventions adapted to their communities' sociocultural needs is vital. Thus, this study seeks to underscore the significance of the COVID-19 pandemic in Acre state by analyzing the epidemiological evolution of COVID-19 from 2020 to 2022.

## 2. Materials and Methods

### 2.1. Study Design and Location

This research follows the design of an ecological time-series study based on a framework outlined by Abreu, Elmusharaf, and Siqueira [17]. Time-series studies allow us to draw valid conclusions from the data by considering the connections between data points that occur over a period [18]. We collected official data from the Brazilian Ministry of Health regarding COVID-19 cases and deaths in Acre, which were publicly available [5].

Acre is located southwest of the Brazilian Amazon basin. It shares borders with Bolivia and Peru and is adjacent to the Brazilian states of Amazonas and Rondônia. Its capital city, Rio Branco, is a hub of commerce and administration and plays a significant role in the region's economic and environmental dynamics (Figure 1). Covering a territorial area of 164,173.429 square kilometers, Acre is home to an estimated population of 830,026 inhabitants as of 2022, with a population density of 5.06 inhabitants per square kilometer. In 2019, the urbanized area of Acre was reported to be 216.14 square kilometers. Regarding socioeconomic indicators, Acre's nominal monthly household income per capita was recorded at BRL 1095 in 2023. The Human Development Index (HDI) for the state, as of 2021, stands at 0.71, reflecting its overall level of development. Healthcare infrastructure in Acre includes 1.9 hospital beds per 1000 inhabitants, with 1.7 hospitalization beds provided by the Unified Health System (SUS) per 1000 inhabitants. The state also has 116 intensive care unit (ICU) beds, of which the SUS provides 81 [11,19].

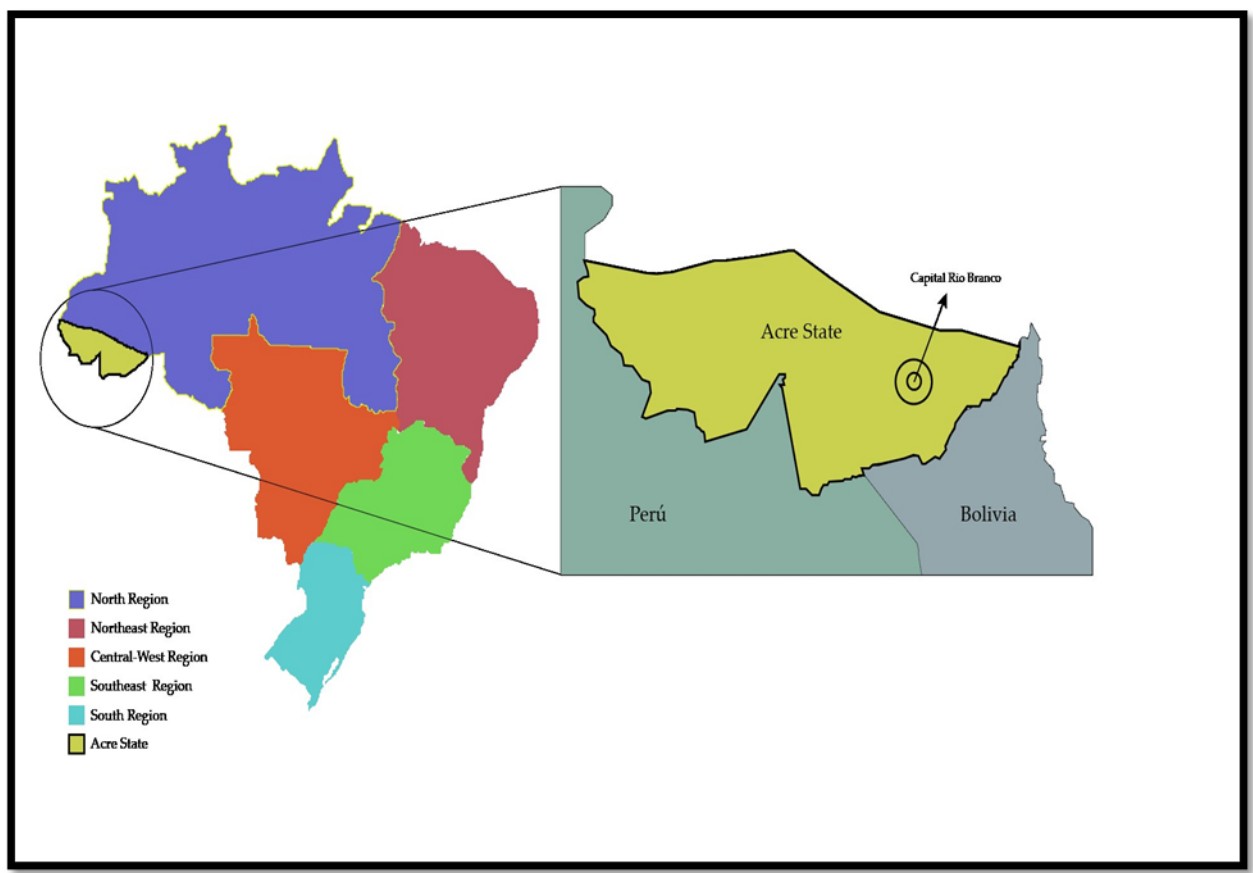

**Figure 1.** Illustrates Acre state's geographical location within the northern region of Brazil, emphasizing its international borders with Peru and Bolivia.

Sample and Eligibility Criteria

We included all COVID-19 cases registered by the Municipal and State Health Departments from March 2020 to December 2022. We also included all deaths confirmed by COVID-19 registered by the Municipal and State Health Departments concerning the previous day. These cases and deaths were confirmed through laboratory tests or clinical and epidemiological confirmation [5]. COVID-19 cases were categorized following the International Classification of Diseases, 10th edition (ICD-10) [20]. To provide context, COVID-19 deaths were defined as those in which the medical certificate included the ICD-10 code B34.2, accompanied by one of the following two codes: U07.1 (confirmed by laboratory testing) or U07.2 (diagnosed clinically or epidemiologically, but without conclusive laboratory tests or unavailable). Deaths due to COVID-19 in Brazil are recorded according to this explanation regardless of the category of the underlying cause of death [5].

We ensured data accuracy by having a second author review the collected information, and a third researcher conducted a final quality check to identify any discrepancies. Finally, we inputted all the data into an Excel spreadsheet (Microsoft Corporation, Redmond, WA, USA) to analyze COVID-19 incidence, mortality rates, and case fatality rates.

### 2.2. Data Analysis and Statistics

We presented the COVID-19 cases and deaths as raw counts (*n*) and percentages relative to the total. We calculated the incidence rate (number of cases per 100,000 people), mortality rate (number of deaths per 100,000 people), and lethality to represent the percentage (%) of people who died among those who contracted COVID-19. We used Acre population estimates from the Brazilian Institute of Geography and Statistics (IBGE) for the years 2020 (866,811 inhabitants), 2021 (878,654 inhabitants), and 2022 (890,220 inhabitants) [11].

Our study aimed to track how COVID-19 indicators evolved over different periods. To achieve this, we analyzed three distinct phases: 2020 (from March to December), 2021 (spanning from January to December), and 2022 (covering January to December). This division allowed us to examine how these indicators changed during these timeframes.

To rigorously review these trends, we followed a well-established protocol by Antunes and Cardoso for conducting time-series studies. Time-series analysis is a method for organizing and making sense of quantitative data collected over time. It is particularly valuable because it enables us to predict past and future values based on statistical calculations, primarily through linear regression [18,21,22].

For our analysis, we constructed time series studies using the Prais–Winsten regression model, a commonly used approach in epidemiological studies. This model was chosen because it accounts for the influence of first-order autocorrelation when examining time-series data. It helps us understand how past data points relate to current ones [18,21–23].

We determined the probability (*p*) and daily percent change (*DPC*) values with a 95% confidence level using specific equations (Equations (1)–(3)). These equations rely on our linear regression analysis's angular coefficient (*β*). The terms "*ul*" and "*ll*" represent the upper and lower limits within the confidence level. These calculations help us assess the statistical significance and quantify the daily percentage change in the data, providing a more precise understanding of our observed trends.

$$DPC = \left(10^{\beta} - 1\right) \times 100\% \tag{1}$$

$$\left(CI\ 95\%\right)_{ul} = \left(10^{\beta_{max}} - 1\right) \times 100\% \tag{2}$$

$$\left(CI\ 95\%\right)_{ll} = \left(10^{\beta_{min}} - 1\right) \times 100\% \tag{3}$$

It is crucial for assessing and categorizing trends in incidence, mortality, and case fatality rates as either increasing, decreasing, or remaining flat. For statistically significant analyses (*p* < 0.05), the daily percentage change (*DPC*) provides insight into the daily variations, indicating whether variables are increasing or decreasing. If the *p*-value is higher than 0.05 (*p* > 0.05), the observed changes in the data are not statistically significant, indicating a stable or unchanging trend. This rigorous statistical approach ensures the accuracy and reliability of our findings and aligns with the specific source or methodology we followed for this statistical analysis [18].

All the statistical analyses were performed using STATA 14.0 software, a commonly used statistical tool (College Station, TX, USA, 2013).

*2.3. Ethical Aspects*

Regarding ethical considerations, the data we used came from reliable information systems maintained by the Ministry of Health. These systems are well established and trusted for analyzing COVID-19 epidemiological indicators. Since the data are publicly available and easily accessible, this study did not need approval from the Scientific Research Ethics Committee.

**3. Results**

Between March 2020 and December 2022, Acre state recorded 158,669 COVID-19 cases and 2040 confirmed deaths. Table 1 illustrates the monthly distribution of these cases and deaths over the entire period under examination.

The first COVID-19 case was detected in March 2020, accounting for 42 cases (0.02%). The first recorded death occurred in April 2020, with 19 deaths (0.97%). In the initial year of the pandemic, June (4.43%) and July (4.01%) stood out with the highest numbers of cases, accompanied by a significant number of deaths, totaling 217 (10.63%) in June and 166 (8.13%) in July.

**Table 1.** Distribution of the absolute and relative frequency of cases and deaths due to COVID-19 per month and year in the state of Acre, Brazil, 2020–2022.

| Year | Month | Cases | | Deaths | |
|------|-------|-------|------|--------|------|
| | | $n^\circ$ | % | $n^\circ$ | % |
| 2020 | March | 42 | 0.02 | - | - |
| | April | 362 | 0.22 | 19 | 0.93 |
| | May | 5815 | 3.66 | 129 | 6.32 |
| | June | 7034 | 4.43 | 217 | 10.63 |
| | July | 6372 | 4.01 | 166 | 8.13 |
| | August | 5022 | 3.16 | 81 | 3.97 |
| | September | 3575 | 2.25 | 47 | 2.30 |
| | October | 2574 | 1.62 | 34 | 1.66 |
| | November | 5463 | 3.44 | 30 | 1.47 |
| | December | 5361 | 3.37 | 72 | 3.52 |
| 2021 | January | 6847 | 4.31 | 72 | 3.52 |
| | February | 9067 | 5.71 | 131 | 6.42 |
| | March | 12,123 | 7.64 | 264 | 12.94 |
| | April | 8146 | 5.13 | 267 | 13.08 |
| | May | 4721 | 2.97 | 133 | 6.51 |
| | June | 3032 | 1.91 | 76 | 3.72 |
| | July | 1585 | 0.99 | 61 | 2.99 |
| | August | 682 | 0.42 | 15 | 0.73 |
| | September | 101 | 0.06 | 24 | 1.17 |
| | October | 126 | 0.07 | 7 | 0.34 |
| | November | 163 | 0.10 | 3 | 0.14 |
| | December | 171 | 0.10 | 3 | 0.14 |
| 2022 | January | 12,876 | 8.11 | 20 | 0.98 |
| | February | 19,323 | 12.17 | 101 | 4.95 |
| | March | 3229 | 2.03 | 20 | 0.98 |
| | April | 1109 | 0.69 | 10 | 0.49 |
| | May | 190 | 0.11 | 0 | 0.00 |
| | June | 1084 | 0.68 | 2 | 0.09 |
| | July | 16,990 | 10.70 | 14 | 0.68 |
| | August | 5864 | 3.69 | 9 | 0.44 |
| | September | 674 | 0.42 | 2 | 0.09 |
| | October | 131 | 0.08 | 0 | 0.00 |
| | November | 4121 | 2.59 | 0 | 0.00 |
| | December | 4694 | 2.95 | 11 | 0.53 |
| Total | | 158,669 | 100.00 | 2040 | 100.00 |

Shifting to the following year (2021), February witnessed the highest number of cases, with 9067 (5.71%), followed by March with 12,123 cases (7.64%). In terms of deaths, March and April recorded the most fatalities, accounting for 264 (12.94%) and 267 (13.98%) deaths, respectively.

By the end of 2022, February (12.17%) and July (10.70%) emerged as the months with the highest COVID-19 case numbers for that year. Notably, February also saw significant deaths, totaling 4.69% of the overall deaths during the entire period under review.

Epidemiological indicators show fluctuations over the entire period analyzed (Figure 2).

Figure 2 shows that the highest incidence rate per 100,000 inhabitants (7859.23) was for 2022. However, the highest mortality rates (120.18 per 100,000 inhabitants) and lethality (2.25%) were reached in 2021.

When analyzing the indicators split by year, we can see that in 2020, the highest incidence and mortality were registered in June, with 811.48 cases (Figure 2a) and 25.03 deaths per 100,000 inhabitants (Figure 2b) while the highest lethality value was observed in April, with 5.24%.

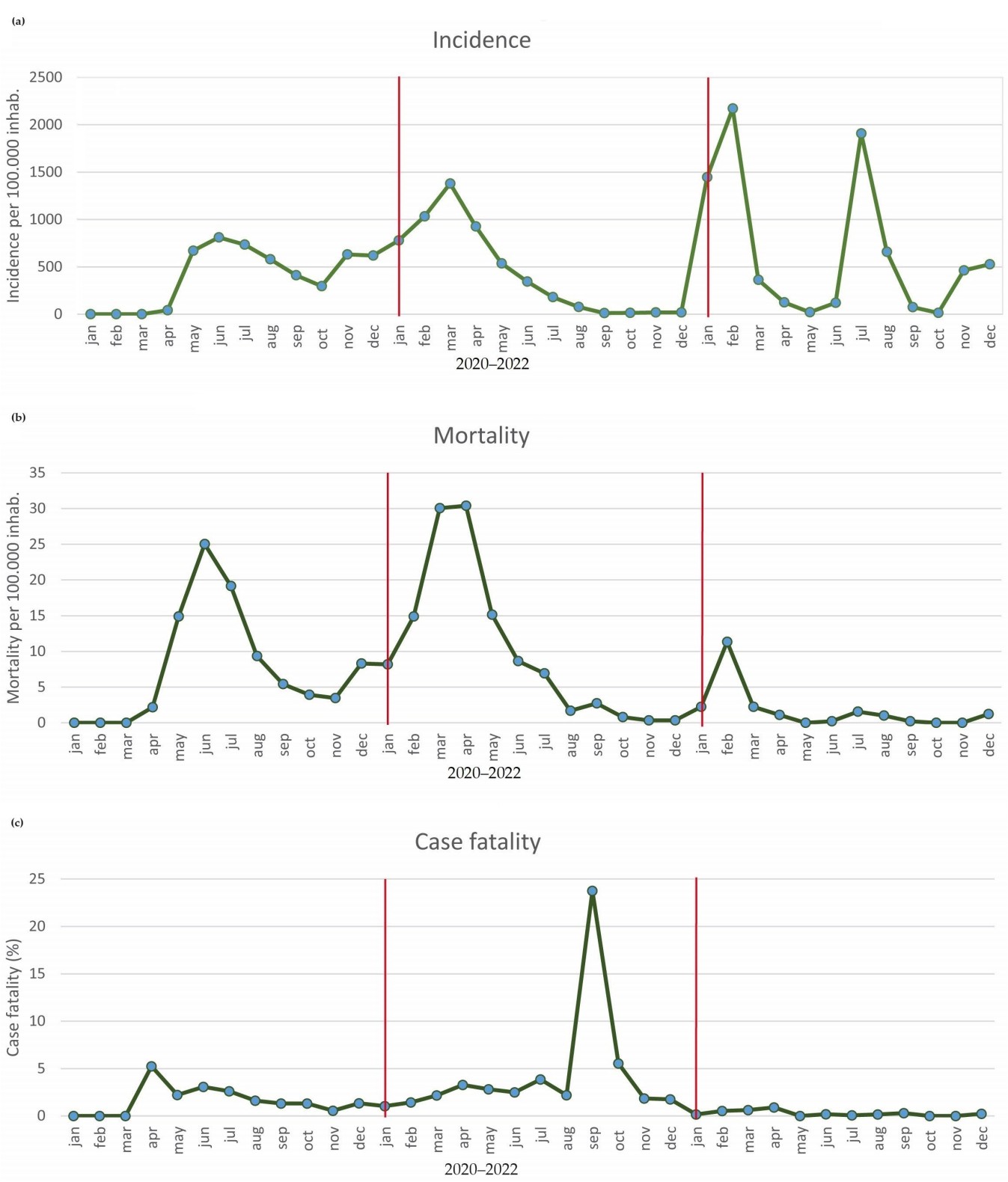

**Figure 2.** Incidence, mortality rates, and lethality of COVID-19 in the state of Acre from January 2020 to December 2022. The temporal analyzes were illustrated as described below: (**a**) incidence, (**b**) mortality and (**c**) case fatality of COVID-19 in the region analyzed, the year was separated by a red bar. Red bars: Separation of the years within the figure.

At the end of 2021, it was observed that March had the highest incidence rate of the year, with 1379.72 cases per 100,000 inhabitants. Mortality was also marked by March,

but it was led by April with 30.04 and 30.38 deaths per 100,000 inhabitants. Lethality peaked in September at 23.76%, showing the highest monthly lethality in the total period analyzed (Figure 2c).

The COVID-19 pandemic exhibits different patterns from 2020 to 2022. Analyzing incidence, mortality, and lethality trends (Table 2) allows for a comprehensive assessment of how the pandemic evolved during the analyzed period and its impact on Acre.

**Table 2.** Prais–Winsten regression estimates and daily percentage change (*DPC*) of incidence, mortality, and lethality rates of COVID-19 in the state of Acre, Brazil, from January 2020 to December 2022.

| Indicators | Year | Linear Regression | | | | |
|---|---|---|---|---|---|---|
| | | *β* | *DPC* | (CI 95%) | *p* | Trend |
| Incidence | 2020 | 0.00434 | 1.00 | 1.00: 0.59 | <0.001 * | Increase |
| | 2021 | −0.00664 | −1.52 | −1.68: −1.35 | <0.001 * | Decrease |
| | 2022 | 0.00057 | 0.13 | −0.31: 0.57 | 0.556 | Flat |
| Mortality | 2020 | −0.00147 | −0.34 | −0.53: −0.14 | 0.001 * | Decrease |
| | 2021 | −0.00212 | −0.49 | −0.76: −0.33 | <0.001 * | Decrease |
| | 2022 | −0.00066 | −0.15 | 0.59: 1.42 | 0.13 | Flat |
| Lethality | 2020 | −0.00238 | −0.55 | −0.55: −0.76 | <0.001 * | Decrease |
| | 2021 | 0.00439 | 1.02 | 1.02: 0.79 | <0.001 * | Increase |
| | 2022 | −0.00176 | −0.40 | −0.40: −1.15 | 0.258 | Flat |

*β*—regression coefficient; *p*—*p*-value; *DPC*—daily percent change; CI 95%—confidence interval 95%. * Statistical difference detected by the Prais–Winsten regression test, $p < 0.05$.

By the close of 2020, the incidence exhibited an upward trend, with a *DPC* of 1%. However, in 2021, there was a notable improvement, and by December's end, it showed a decreasing trend, with a *DPC* of −1.52%. Subsequently, in 2022, there was a significant surge in cases, resulting in a stable incidence trend by the end of the fourth quarter.

Mortality displayed decreasing trends for 2020 and 2021, with *DPC* values of −0.34 and −0.49, respectively. However, toward the end of 2022, a stationary trend emerged.

Lethality, on the other hand, followed a different pattern. It decreased toward the end of 2020, with a *DPC* of −0.50. However, it had a negative impact in 2021, leading to a growing trend by the end of the year, indicated by a daily percentage variation of 1.02%. In 2022, the situation improved, resulting in a stable lethality trend by the end of the year.

## 4. Discussion

Throughout the examination period, Acre witnessed significant fluctuations in COVID-19 cases and fatalities, manifesting in variations across incidence, mortality, and lethality rates. Notably, by late April 2020, the state encountered heightened community transmission of the coronavirus [24]. Despite proactive measures and advantages, such as a well-equipped infectious disease laboratory in the capital facilitating early case detection, viral spread persisted. Prompt enforcement of interventions, including school closures and restrictions on non-essential activities, aimed to alleviate strain on the healthcare system. The Ministry of Health advocated for personal and public hygiene practices, including regular handwashing and sanitization. Additionally, they recommended isolating individuals experiencing symptoms for 14 days and emphasized the importance of using personal protective equipment by both patients and healthcare professionals [25]. Nonetheless, despite these efforts, fatalities remained a grim reality [26].

It is noteworthy that Acre, along with Roraima and Amapá, neighboring states in the northern region of Brazil, exhibited low hospitalization rates by COVID-19 (0.1%, 0.8%, and 0.3%, respectively) but disproportionately high hospital death rates (43.8%, 35.9%, and 44.7%). This indicates a lack of capacity or minimal resources to manage severe cases in these regions [27] effectively. The northeast region showed a higher hospital mortality rate among non-elderly patients at 31%, compared to a lower rate of 15% in the country's southern region [28].

Data from the IBGE reveal that states in the northern and northeastern regions are comparatively under-equipped with mechanical ventilation equipment. For instance, Amapá, Piauí, Maranhão, Alagoas, and Acre exhibit low indices of ventilators per 100 thousand inhabitants, ranging from 10 to 16. Additionally, the availability of intensive care beds per 100,000 inhabitants in Acre, Amapá, and Roraima is among the lowest in the country, with rates as low as 4 to 5 ventilators per 100,000 inhabitants [29].

During the initial year of the pandemic (2020), June and July saw a surge in COVID-19 cases and fatalities. Notably, this period coincided with the onset of winter, characterized by drier weather and lower temperatures due to the influence of the Atlantic Polar Mass in the region [30]. These conditions often lead to sudden drops in temperature, creating a colder environment. Additionally, winter is typically associated with increased respiratory illnesses, exacerbated by the dry and cold air irritating the airways [31]. Consequently, these factors likely contributed to a higher prevalence of respiratory symptoms related to COVID-19, resulting in elevated case numbers and mortality rates during this period. Like other regions, Acre has experienced the circulation of various virus variants. During the analyzed period, four distinct waves in the behavior of epidemiological indicators were observed in Acre. The first wave occurred between March and December 2020, mainly driven by lineages B.1.1.28 and B.1.1.33, which probably emerged in February 2020 and were the most prevalent variants until October 2020. The second wave occurred during the first two quarters of 2021, coinciding with the dominance of the variant of concern (VOC)-Gamma. The third wave occurred between June and August 2021, corresponding to the predominance of the VOC-Delta. The final wave emerged at the end of 2021, with the appearance of the VOC-Omicron [32,33].

In 2021, February and March saw a peak in case numbers, with March and April experiencing the highest number of deaths in Acre. The VOC-Gamma was initially identified in December 2020 and quickly became predominant in Acre and throughout Brazil, accounting for over 96% of reported cases from January to June. This variant has been linked to heightened transmission rates and increased disease severity. Furthermore, it has been associated with diminished efficacy of both treatments and vaccines. After July, the transmission landscape shifted from the VOC-Gamma to the VOC-Delta. Despite the rapid spread of the VOC-Delta, there was no significant increase in reported cases and deaths in Acre. This outcome can be attributed to the effectiveness of the early vaccination campaign and the natural immunity acquired from prior infection with the Gamma variant [34]. In December 2021, 39.4% of sequenced genomes were attributed to the VOC-Omicron, marking a substantial presence. By January 2022, this percentage surged to 95.9%, signifying rapid and widespread variant dissemination [35]. The VOC-Omicron was identified in all regions and became the dominant force shaping the epidemiological landscape of COVID-19 in Brazil [35,36].

February and July of 2022 emerged as the months with the highest recorded cases of COVID-19. Notably, February stood out with the highest case count observed throughout the analyzed pandemic. The surge in COVID-19 cases and the notable increase in incidence rates at the onset of this year align with findings from the Fiocruz observatory report, which attributes the outbreak to the BA variant. This variant, particularly the BA.2 strain, spread across Brazil and other nations alongside the BA.4 and BA.5 variants, exhibiting faster transmission rates than preceding mutations before VOC-Omicron [36,37]. Despite the exponential rise in cases and incidence rates of COVID-19 in Acre during January and February 2022, the fatalities did not escalate in the same proportion. This discrepancy suggests potential factors at play, such as improvements in treatment protocols or changes in the virus's virulence or pathogenicity.

The findings showed that the highest COVID-19 mortality rate in the three years of the pandemic was identified in 2021, the initial moment of mass vaccination [38]. Variations in the COVID-19 mortality rate by age in the capitals of all Brazilian states showed that the highest crude mortality rate was found in Manaus, a city in Amazonia in the country's northern region. Furthermore, the age-standardized rate increased in all capitals in the

north area, including Rio Branco, the capital of the state of Acre [31]. According to research conducted by the Getulio Vargas Foundation [39], Acre faces significant challenges in providing healthcare access to its population, with 45.53% living on a monthly income of fewer than BRL 497 (less than USD 100). This disparity in access to resources exacerbates the susceptibility of low-income populations to COVID-19. Factors such as reliance on public transportation, higher household densities, limited access to basic sanitation and healthcare, and the struggle to maintain social isolation due to income loss or job insecurity all contribute to a heightened risk of infection [40].

A retrospective cohort study in Acre identified various risk factors associated with COVID-19-related mortality. These factors encompass male gender, age over 60 years, dyspnea, presence of multiple comorbidities, reporting of sore throat and headache, and being a healthcare professional [26]. Interestingly, men have expressed concerns about adverse reactions as their primary reason for vaccine hesitancy. This highlights potential implications for their mortality risk [41]. In contrast, a separate study investigated the impact of social isolation measures prompted by COVID-19 on the quality of life (QoL) in two distinct locations: Rio Branco, Acre, and Santo André, São Paulo. Surprisingly, those residing in Rio Branco demonstrated a better QoL than those in Santo André. However, both locations experienced compromised QoL as a result of social isolation measures [42].

In 2020, we observed an increasing trend in the incidence of COVID-19, accompanied by a simultaneous decrease in mortality and lethality rates. However, lethality rates rose while incidence and mortality decreased the following year. Notably, the highest percentage of lethality was observed in September 2021. This discrepancy in lethality rates across different months can be attributed to several factors influencing the calculation of these rates. Fluctuations in testing capacity and delays in reporting flow significantly impact the reported number of confirmed COVID-19 cases [43]. During months with widespread testing efforts or improved accessibility to testing facilities, a more significant proportion of asymptomatic or mild cases may be detected, contributing to a lower overall lethality rate. Conversely, during periods of limited testing availability or when testing is primarily focused on symptomatic individuals, the reported cases may predominantly consist of severe or critical cases, resulting in a higher observed lethality rate.

The transition from an increasing to a decreasing trend in COVID-19 incidence from 2020 to 2021 may have been positively influenced by initiating vaccination campaigns and reinforcing non-pharmacological measures, such as social distancing and mask mandates, within the Acre population [44]. Despite these efforts, national data from Fiocruz's COVID-19 Observatory Bulletin indicated persistent high incidences of severe acute respiratory syndromes (SARS) across Brazil in early 2021, with Acre experiencing a particularly elevated moving average incidence [45]. In response, the COVID-19 vaccination campaign was launched in mid-January 2021, representing a crucial moment in Brazil's battle against the pandemic [38]. All federative units, including Acre, initially prioritized individuals at high risk, such as healthcare workers, older adults, people with chronic diseases, the homeless, and indigenous people, before gradually expanding to the entire population based on decreasing age. The initial vaccines administered were CoronaVac (Sinovac Biotech, Beijing, China) and ChAdOx1 nCoV-19 (AstraZeneca/Oxford University, Oxford, United Kingdom), followed by BNT162b2 (Pfizer-BioNTech, Mainz, Germany) and Ad26.Cov2.S (Johnson & Johnson-Janssen, Leiden, The Netherlands) in May and June 2021 [46].

Despite extensive efforts during the vaccination campaign, Acre had the lowest vaccination coverage among all Brazilian states by the third quarter of 2021 [47]. Notably, according to the Fiocruz report from October 2021, the effectiveness of these vaccines in Brazil was high. All vaccines demonstrated significant efficacy in reducing the risk of infection, hospitalization, and death from COVID-19. Among individuals aged 20 to 80, the protection offered by all four vaccines ranged from 83% to 99%. Similarly, in the population under 60, all vaccines showed robust protection, with efficacy rates exceeding 85% against the risk of hospitalization and surpassing 89% against the risk of death [48]. Nevertheless, opinions arose regarding the enduring efficacy of the vaccines, spurred by

the emergence of new variants and a rise in documented infections. As in the rest of Brazil, there was considerable anticipation surrounding the efficacy of COVID-19 vaccines in Acre. Emerging evidence suggests their protection against fatalities surpassed their effectiveness against severe cases. Initial data indicate that the first immunization series conferred a substantial level of protection (>50%) for up to approximately 130 days (or 19 weeks) for most vaccines, excluding Coronavac in the elderly population. Notably, booster doses of ChAdOx1 nCoV-19 and BNT162b2 vaccines demonstrated enhanced protection compared to the first immunization series [49].

Research has shown the effectiveness of COVID-19 vaccines in averting severe illness, hospitalization, and mortality [48,49]. In Acre, as in other regions, the vaccination campaign has significantly bolstered immunity across the population, thereby alleviating the virus's impact on public health outcomes. This is evidenced by the decrease in mortality and lethality rates in 2022. While other factors, such as variant emergence and adherence to preventive measures, also influence death rates, the data consistently highlight the positive impact of vaccination in reducing mortality associated with COVID-19 in Acre.

The state of Acre presents a complex scenario influenced by multiple variables affecting the pandemic and the spread of COVID-19 [7,50], leading to numerous challenges impacting incidence, mortality, and lethality rates. Among its 22 municipalities, 14 are categorized as rural [11], where rivers serve as crucial transportation routes for people and goods, and in some areas, they represent the only form of access. Alongside deficiencies in access to basic sanitation, it has emerged as a significant issue for the state's infrastructure and social sectors. In 2022, over half of Acre's population (69.5%) did not receive regular access to water, and 13 out of every 100 individuals lived in households lacking exclusive bathroom facilities [51].

The pandemic revealed the scarcity of healthcare resources and laid bare deep inequalities, not only between public and private healthcare, but also within SUS itself and in the difference in the supply of ICU beds between municipalities and regions in the country [52–55]. In this way, the three-year pandemic scenario in the state of Acre presented a combination of factors that favored the transmission of the virus and justified the high number of cases and deaths.

After thoroughly discussing our study's findings, it is essential to address its limitations. Despite the valuable insights gained, our research is not without its constraints. While secondary data analysis offers advantages through the utilization of existing information, it can also be subject to significant limitations, often stemming from the integrity and depth of the selected dataset. A critical concern is the potential incompleteness of detail within the underlying data. Additionally, the absence of individual variables such as age, sex, and race can substantially constrain the study's conclusions, diminishing the robustness and generalizability of the results obtained. It is important to note that the registration of cases and deaths may be underestimated relative to the actual number due to restrictions on conducting widespread COVID-19 testing during the analyzed three-year period. However, this study provides a comprehensive representation of the pandemic trajectory in the state of Acre, which has limited visibility in epidemiological behavior studies during the COVID-19 pandemic.

## 5. Conclusions

During the analyzed period, COVID-19 spread in Acre and showed different patterns over time. COVID-19 cases increased, with waves of higher and lower infection rates. But 2021 was the most fatal year; it had the highest fatality rate of the entire period studied (24.76% in September) and a daily lethality growth rate of 1.02%. Although COVID-19 presented stationary incidence, mortality, and lethality rates at the end of 2022, constant epidemiological disease monitoring is necessary. Indeed, continuous monitoring of the most predominant SARS-CoV-2 lineages and their dynamics is imperative. Such proactive actions are essential for addressing emerging challenges and ensuring effective responses to adverse situations.

This study emphasizes the importance of continued research and vigilance in regions like Amazonia, where healthcare vulnerabilities persist, and infectious diseases remain a significant concern.

**Author Contributions:** Conceptualization: A.R.P.R., B.E.G.D., J.E.T.M., T.C.M. and L.C.d.A.; methodology: B.E.G.D., T.C.M. and L.C.d.A.; software: B.E.G.D., T.C.M. and L.C.d.A.; validation: A.R.P.R., B.E.G.D., T.C.M. and L.C.d.A.; formal analysis: B.E.G.D., L.C.d.A., M.P.E.C. and T.C.M.; investigation: A.R.P.R., J.E.T.M., I.M.P.B. and M.P.E.C.; resources: J.E.T.M. and L.C.d.A.; data curation: B.E.G.D., J.E.T.M., M.P.E.C. and T.C.M.; writing—original draft preparation: B.E.G.D., J.E.T.M., I.M.P.B., M.N., M.P.E.C. and T.C.M.; writing—review and editing: A.R.P.R., B.E.G.D., J.E.T.M., I.M.P.B., L.C.d.A., M.N., M.P.E.C. and T.C.M.; visualization: B.E.G.D., M.P.E.C. and T.C.M.; supervision: A.R.P.R. and L.C.d.A.; project administration: A.R.P.R. and L.C.d.A. All authors have read and agreed to the published version of the manuscript.

**Funding:** This research received no external funding.

**Institutional Review Board Statement:** Not applicable.

**Informed Consent Statement:** Not applicable.

**Data Availability Statement:** Data were extracted from: https://covid.saude.gov.br/ (accessed on 23 January 2023).

**Acknowledgments:** The authors extend their gratitude to the Instituto Federal Goiano for the support received for the dissemination of the results of this study.

**Conflicts of Interest:** The authors declare no conflicts of interest.

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
