# Peer review of "Analyzing the COVID-19 Transmission Dynamics in Acre, Brazil: An Ecological Study"

_epidemiologia, doi:10.3390/epidemiologia5020013_

Round 1
Reviewer 1 Report
Comments and Suggestions for Authors
Please see the enclosed sheet. Major revision is required.

English composition is poor, some paragraphs have just 3 lines.
Author Response
Dear reviewer 1, thank you very much for your comments.

Reviewer 2 Report
Comments and Suggestions for Authors
Thank you for sharing your article assessing the impact and persistence of COVID-19 in a Brazilian setting by applying a time series analysis. The following comments may help to improve the article:
L79: Please explain to potential readers how the reported cases and deaths were confirmed.
Table 1: Please rephrase "populacao estimada" to population estimated.
L89: Were the deaths reported due to COVID or any other underlying reason?
L92-93: If the date does not reflect the date of death, what else was confirmed by laboratory and clinical data on that date?
L109: Please add the source(s) for the population estimates. Do they relate only to Acre State?
General comment: The manuscript would benefit from presenting also the viral variants present during the three observational periods. Please consider adding those data should they be available. Also, please add throughout more information on the mass vaccination administered, including the vaccine(s) given, the frequency of administration and the groups/age groups targeted. In terms of infectious control and prevention, please also outline the anti-COVIS-19 measures that were administered during the study period. As ready stated, please provide more information on the laboratory and clinical/epidemiological confirmation of cases and deaths, including whether the deaths reported occurred solely due to COVID-19.
Author Response
Dear Reviewer 2, thank you very much for your comments.

Round 2
Reviewer 1 Report
Comments and Suggestions for Authors
Paper has been improved based on the first round, still need major revision.
Please see the PDF sheet enclosed.

Needs improvement
Author Response
ROUND 2
Manuscript: Epidemiologia 2953911
Ref.: Responses to Reviewer 1
Dear Reviewer 1:
Thank you for providing further feedback on the points you find critical. In this response, we aim to address your additional points and clarify where necessary. We appreciate your insights into ensuring the quality and rigor of our work to strengthen our manuscript.
The authors inform that in the manuscript the text highlighted in yellow are the changes that were made previously, and in green are the changes made in Round 2.
See the responses to your remarks on Round 2 as follows:
- Comment #1:
In my first round report, I gave some suggestions to improve the paper. The paper has improved to some extent, still needs revision. In second round of revision authors must write point-by-point answers to my comments. I can not see much change they made.
Response:
We understand your concern regarding the need for further revisions and the importance of explicitly addressing your suggestions. We apologize if our response in the first round did not sufficiently meet your expectations. In this second round of revision, we assure you that we will provide a thorough, point-by-point response to each of your comments. We acknowledge the necessity of clearly demonstrating the changes made in response to your feedback.
- Comment #2
Line 65-68, just 4 lines paragraph- merge with other paragraphs (short paragraphs we do not see in any writings, newspapers, magazines research articles etc.). Similarly merge a few lines paragraph appearing at other places in the paper to other paragraph to make paper up-to-date.
Response:
We want to assure you that we have carefully considered your suggestion and have implemented the necessary changes throughout the manuscript. In response to your feedback, we have merged the short paragraphs in lines with adjacent paragraphs to create a more cohesive narrative structure (Line 65-76). Additionally, we have identified and merged other brief paragraphs throughout the paper to enhance readability.
- Comment # 3:
Include a map that shows the location of the state of Acre in Brazil map (as indented the location in Table 1).
Response:
We agree that incorporating a map of Acre State will significantly benefit our readers by visually representing the geographic context in which our study was conducted (line: 98).
- Comment # 4:
Exclude the details in the table which has no relation with the paper title or it is not related to the SARS-CoV-2 spread. The level of the table contents is very low at the primary school it can not be published in MDPI journals. MDPI journals are of high impact factor journals. Reviewers and Editors must maintain the level of the research i.e. novelty.
Response:
We understand your concern about the level of detail provided in Table 1. At the same time, it may appear elementary at first glance. Still, the sociodemographic characteristics of the population in Acre State play a crucial role in understanding the dynamics of COVID-19 mortality in this region. Given that our paper is intended for an international audience, including readers who may not be familiar with the nuances of Acre State, we believe that providing this foundational sociodemographic information is important to enable a comprehensive understanding of the epidemiological landscape.
However, in response to your comment, we reformulated the data and presented it within the text of the paper. The text is inserted in lines 84-96 as follows:
Acre is located southwest of the Brazilian Amazon basin. It shares borders with Bolivia and Peru and is adjacent to the Brazilian states of Amazonas and Rondônia. Its capital city, Rio Branco, is a hub of commerce and administration and plays a significant role in the region's economic and environmental dynamics. Covering a territorial area of 164,173.429 square kilometers, Acre is home to an estimated population of 830,026 inhabitants as of 2022, with a population density of 5.06 inhabitants per square kilometer. In 2019, the urbanized area of Acre was reported to be 216.14 square kilometers. Regarding socioeconomic indicators, Acre's nominal monthly household income per capita was recorded at 1095 Brazilian Reais in 2023. The Human Development Index (HDI) for the state, as of 2021, stands at 0.71, reflecting its overall level of development. Healthcare infrastructure in Acre includes 1.9 hospital beds per 1,000 inhabitants, with 1.7 hospitalization beds provided by the Unified Health System (SUS) per 1,000 inhabitants. The state also has 116 ICU beds, of which the SUS provides 81 (11,19)
- Comment #5:
In plot Figure 1, what is /AC? I can see per 100,000 population mentioned in the paper.
Response:
As it pertains to Brazilian databases, we have removed the identification 'AC' for Acre from the figures included in this paper.
Comment # 6.
6.1: The waves in the Figure 1, can be identified by Wuhan, Delta, Omicron etc.
6.2: In Sept 2021, why fatalities % are so high? Mortalities must also be higher in this month or time range. Which are not. Number of cases i.e. infections are proportional to the fatalities and % deaths which I can not see in the Figure 1, no relationship?
Responses:
Response to 6.1:
While we understand the importance of considering COVID variants in understanding mortality trends, we have chosen not to modify the graphs as initially presented. The decision was made to maintain the clarity and focus of each graph on specific metrics (incidence, mortality, and fatality) without introducing additional variables that could potentially complicate interpretation. However, in the Discussion section, we have provided a detailed sequence of COVID variants observed, which provides context for understanding the evolving landscape of the pandemic in Acre. We believe that this approach offers a comprehensive understanding of the factors contributing to mortality trends without altering the presentation of the primary data.
This is explained in the section Discussion as follows: (Line 293-327)
Like other regions, Acre has experienced the circulation of various virus variants. Dur-ing the analyzed period, four distinct waves in the behavior of epidemiological indicators were observed in Acre. The first wave occurred between March and December 2020, mainly driven by lineages B.1.1.28 and B.1.1.33, which probably emerged in February 2020 and were the most prevalent variants until October 2020. The second wave occurred during the first two quarters of 2021, coinciding with the dominance of the Variant of concern (VOC) Gamma. The third wave occurred between June and August 2021, corre-sponding to the predominance of the VOC-Delta. The final wave emerged at the end of 2021 with the appearance of the VOC-Omicron [32,33].
In 2021, February and March saw a peak in case numbers, with March and April ex-periencing the highest number of deaths in Acre. The VOC-Gamma was initially identi-fied in December 2020 and quickly became predominant in Acre and throughout Brazil, accounting for over 96% of reported cases from January to June. This variant has been linked to heightened transmission rates and increased disease severity. Furthermore, it has been associated with diminished efficacy of both treatments and vaccines. After July, the transmission landscape shifted from the VOC-Gamma to the VOC-Delta. Despite the rapid spread of VOC-Delta, there was no significant increase in reported cases and deaths in Acre. This outcome can be attributed to the effectiveness of the early vaccination cam-paign and the natural immunity acquired from prior infection with the Gamma variant [34]. In December 2021, 39.4% of sequenced genomes were attributed to the VOC-Omicron , marking a substantial presence. By January 2022, this percentage surged to 95.9%, signifying rapid and widespread variant dissemination. The VOC-Omicron was identified in all regions and became the dominant force shaping the epidemiological landscape of COVID-19 in Brazil [36].
February and July of 2022 emerged as the months with the highest recorded cases of COVID-19. Notably, February stood out with the highest case count observed through-out the analyzed pandemic. The surge in COVID-19 cases and the notable increase in in-cidence rates at the onset of this year align with findings from the Fiocruz observatory re-port, which attributes the outbreak to the BA variant. This variant, particularly the BA.2 strain, spread across Brazil and other nations alongside the BA.4 and BA.5 variants, ex-hibiting faster transmission rates than preceding mutations before VOC-Omicron [36,37]. Despite the exponential rise in cases and incidence rates of COVID-19 in Acre during January and February 2022, the fatalities did not escalate in the same proportion. This discrepancy suggests potential factors at play, such as improvements in treatment protocols or changes in the virus's virulence or pathogenicity
Response to 6.2:
The elevated fatality percentage in September 2021 can be attributed to the relatively high number of deaths compared to the number of reported cases during that month. Specifically, in September 2021, 101 cases and 24 deaths were reported in our dataset. It results in an approximate fatality rate of nearly 24%. It's important to note that Figure 1 illustrates lethality as a percentage relative to the number of reported cases, as outlined in the method section.
An explanation related to the high lethality of September is included in the section Discussion lines: 352-363 as follows:
However, lethality rates rose while incidence and mortality decreased the following year. Notably, the highest percentage of lethality was observed in September 2021. This discrepancy in lethality rates across different months can be attributed to several factors influencing the calculation of these rates. Fluctuations in testing capacity and delays in reporting flow significantly impact the reported number of confirmed COVID-19 cases. During months with widespread testing efforts or improved accessibility to testing facilities, a more significant proportion of asymptomatic or mild cases may be detected, contributing to a lower overall lethality rate. Conversely, during periods of limited testing availability or when testing is primarily focused on symptomatic individuals, the reported cases may predominantly consist of severe or critical cases, resulting in a higher observed lethality rate.
- Comment #7:
Vaccination has been discussed, what is the effect of vaccination on the death rates in Acre. Paper still needs major revision.
Response:
The following text is in the discussion section Lines 369-401
In response, the COVID-19 vaccination campaign was launched in mid-January 2021, representing a crucial moment in Brazil's battle against the pandemic [38]. All federative units, including Acre, initially prioritized individuals at high risk, such as healthcare workers, older adults, people with chronic diseases, the homeless, and indigenous people, before gradually expanding to the entire population based on decreasing age. The initial vaccines administered were CoronaVac (Sinovac Biotech) and ChAdOx1 nCoV-19 (AstraZeneca/Oxford University), followed by BNT162b2 (Pfizer-BioNTech) and Ad26.Cov2.S (Johnson & Johnson-Janssen) in May and June 2021 [46].
Despite extensive efforts during the vaccination campaign, Acre had the lowest vac-cination coverage among all Brazilian states by the third quarter of 2021 [47]. Notably, ac-cording to the Fiocruz report from October 2021, the effectiveness of these vaccines in Bra-zil was high. All vaccines demonstrated significant efficacy in reducing the risk of infec-tion, hospitalization, and death from COVID-19. Among individuals aged 20 to 80, the protection offered by all four vaccines ranged from 83% to 99%. Similarly, in the popula-tion under 60, all vaccines showed robust protection, with efficacy rates exceeding 85% against the risk of hospitalization and surpassing 89% against the risk of death [48]. Nev-ertheless, opinions arose regarding the enduring efficacy of the vaccines, spurred by the emergence of new variants and a rise in documented infections. As in the rest of Brazil, there was considerable anticipation surrounding the efficacy of COVID-19 vaccines in Acre. Emerging evidence suggests their protection against fatalities surpassed their effec-tiveness against severe cases. Initial data indicate that the first immunization series con-ferred a substantial level of protection (>50%) for up to approximately 130 days (or 19 weeks) for most vaccines, excluding Coronavac in the elderly population. Notably, booster doses of ChAdOx1 nCoV-19 and BNT162b2 vaccines demonstrated enhanced protection compared to the first immunization series [49].
Research has shown the effectiveness of COVID-19 vaccines in averting severe illness, hospitalization, and mortality [48,49]. In Acre, as in other regions, the vaccination cam-paign has significantly bolstered immunity across the population, thereby alleviating the virus's impact on public health outcomes. This is evidenced by the decrease in mortality and lethality rates in 2022. While other factors, such as variant emergence and adherence to preventive measures, also influence death rates, the data consistently highlight the pos-itive impact of vaccination in reducing mortality associated with COVID-19 in Acre.

Reviewer 2 Report
Comments and Suggestions for Authors
Thank you for sharing the revised manuscript. All my comments were addressed sufficiently.
Author Response
Dear Reviewer 2, thank you for your evaluation, your notes were fundamental for improving this work. Some modifications were made, as requested by the other reviewer. We are happy for your review. The authors thank you for your collaboration.